



# Extreme precipitation events in the Mediterranean area: contrasting Lagrangian and Eulerian models for moisture sources identification

Sara Cloux [1], Daniel Garaboa-Paz [1], Damián Insua-Costa [1], Gonzalo Miguez-Macho [1], and Vicente Pérez-Muñuzuri [1]

[1]CRETUS Institute, Group of Nonlinear Physics, Faculty of Physics, University of Santiago de Compostela, Spain

**Correspondence:** Sara Cloux (s.cloux@usc.es)

**Abstract.**

Concern about heavy precipitation events has increasingly grown in the last years in Southern Europe, especially in the Mediterranean region. These occasional episodes can result in more than 200 mm of rainfall in less than 24 h, producing flash floods with very high social and economic losses. To the better understanding of this phenomena, a correct identification of

the origin of the moisture must be found. However, the contribution of the different sources is very difficult to estimate from observational data, so numerical models are usually used to this end. Here, we present a comparison between two complex methodologies for the quantification of the moisture sources in two infamous events occurred during October and November 1982 in the Western Mediterranean area. In a previous study, using an Eulerian approach it was determined the contributions of moisture evaporated in: 1) Western Mediterranean; 2) Central Mediterranean; 3) North Atlantic ocean and 4) tropical and

subtropical Atlantic and tropical Africa. Now, we use the Lagrangian model FLEXPART-WRF to quantify the role played by these sources. Considering the results provided by the Eulerian analysis as the "virtual reality", we validated the performance of the Lagrangian model. Results show that the Lagrangian method has an acceptable performance in identifying local (Western Mediterranean) and medium-distance (Central Mediterranean and North Atlantic) sources. However, remote moisture sources, like tropical and subtropical areas, are underestimated by the Lagrangian approach. Notably, for the October event,

the tropical and subtropical area reported a relative contribution six times below than the Eulerian method. In contrast, the FLEXPART-WRF overestimates the contribution of some sources, especially from the Sahara. We argue that such an inconsistent contribution is associated with the fact that the Lagrangian method does not consider moisture phase changes. These over- and underestimates should be taken into account by other authors when drawing conclusions from the Lagrangian analysis.

## 1 Introduction

Extreme rainfall and associated floods are one of the most devastating agents in the weather context. As an example, in 2019 alone, more than 5000 people died as a result of floods and more than 30 million people were affected, resulting in economic losses of more than USD 35 billion (International Disaster Database; dat). Therefore, the study of such catastrophic events is fundamental for prediction and anticipation.





Heavy precipitation events (HPEs) are distributed unevenly across the planet, and there are places that are very prone to receiving enormous rain accumulations in a short time, as opposed to other areas where the precipitation regime is much more moderate. One of these regions where extreme precipitation and flooding are very recurring is the Western Mediterranean Region (WMR) (Llasat et al., 2010). Several exemptions make this part of the planet so exposed to heavy precipitation; the Mediterranean Sea being a large and mild water body, enclosed by very complex orography and in a relatively northern latitude are some examples (e.g., Buzzi et al., 1998; Llasat, 2009; Dayan et al., 2015). In this area, most of the events take place in autumn (Mariotti et al., 2002), when Atlantic lows or cut-off lows (Nieto et al., 2005) often interact with warm Mediterranean Sea waters leading to strong convection.

In spite of HPEs are a regional phenomena, moisture feeding them not only comes from nearby sea evaporation, but can originate in remote regions and be transported by different atmospheric mechanisms. In this sense, it has been shown that long-distance moisture transport through atmospheric rivers (ARs) is a crucial contributor to total precipitation amounts recorded in Europe and the United States (Lavers and Villarini, 2015) and also to extreme rainfall episodes (e.g., Stohl et al., 2008; Eiras-Barca et al., 2017). As for the WMR, recent studies (e.g., Winschall et al., 2012; Pinto et al., 2013; Krichak et al., 2015; Insua-Costa et al., 2018) suggest that remote sources of moisture such as the North Atlantic or tropical or subtropical areas could contribute significantly to the frequent torrential rains there.

In order to find the origin of the moisture, different methodologies have been used in the past (see Gimeno et al. (2012), for a detailed review of numerical methods used in moisture source studies), being the Lagrangian models the most widely used technique. It is based on the analysis of the moisture content change of air parcels being tracked backward (or, less commonly, forward) in time. Lagrangian methods are offline, and therefore very efficient from a computational point of view. On the other hand, Eulerian-type (online) methods are much more computationally expensive, and therefore have been less used. However, it is considered to be the most accurate tool for moisture sources studies. In the western Mediterranean, the Lagrangian methods have been used by several authors (e.g., Reale et al., 2001; Turato et al., 2004; Nieto et al., 2010; Duffourg and Ducrocq, 2011), while the Eulerian approach has only been used by Winschall et al. (2012) and Insua-Costa et al. (2018).

In this study, we analyzed the moisture sources in two catastrophic flooding episodes occurred in the WMR, using both the Lagrangian and Eulerian approaches. The Eulerian model for moisture tracking employed was the WRF-WVTs (Insua-Costa and Miguez-Macho (2018)). The results obtained using this tool has already been presented in a previous article (Insua-Costa et al., 2018). Thus, the objective of this study is to repeat the same strategy, but in this case using the Lagrangian FLEXPART-WRF model (Brioude et al., 2013), so that we can intercompare the results provided by both methodologies. This type of comparison has already been presented recently by Winschall et al. (2014). However, the present study is especially focused on identifying the possible limitations of the Lagrangian method and their causes, based on the assumption that WRF-WVTs represent the "virtual reality". This was the same strategy followed by van der Ent et al. (2013), where the evaporated moisture from Lake Volta (in West Africa) was tracked until it precipitates, concluding that the Lagrangian method leads to inaccuracies





in the calculations in the presence of strong wind shear. One of the novelties with respect to this article, is that here the sources of moisture will be analyzed from a non-local point of view, that is to say, a large scale domain has been employed to be able to cover the moisture source of remote origin. The aim is to check whether the Lagrangian method has the same capacity to detect short-distance sources as long-distance ones. In summary, the present work is intended to contribute to improving the Lagrangian analysis of moisture sources on the basis of another technique (WRF-WVTs) that is more accurate but much more

costly from the computational point of view and therefore less practical.

This study is structured as follows: in Section 2, Lagrangian and Eulerian methodologies are described and the procedure presented. The two case studies are briefly introduced in the first part of Subsections 3.1.1 and 3.1.2 respectively. More detailed description can be found in Insua-Costa et al. (2018). After that, results are structured as follows: Section 3.1 presents the mois-

ture sources analysis obtained from the Lagrangian method, Section 3.2 shows the comparison between the FLEXPART-WRF and WRF-WVTs techniques and Section 3.3 discusses the limitations of the Lagrangian approach. Finally, Section 4 summarizes and concludes this work.

## 2 Methods

### 2.1 Lagrangian approach

In this study we use the FLEXPART model (FLEXible PARTicle dispersion model; Stohl and James (2004)), which has been widely used to study moisture sources from a climatological perspective (e.g. James et al., 2003; Ciric et al., 2018; Drumond et al., 2014; Gimeno et al., 2013) as well as in particular heavy precipitation events (e.g. Stohl et al., 2008; Sun and Wang, 2014) . Specifically, we use a version of FLEXPART that works with the Weather Research and Forecasting (WRF) regional

atmospheric model (et al. Skamarock WC (2008)), known as FLEXPART-WRF (Brioude et al. (2013)).

According to the distribution of atmospheric mass, the simulation domain (covering from $90°$W to $60°$E and from $3°$S to $65°$N, Figure 1 (a)) is homogeneously divided into 4 million air parcels (or particles), which are subsequently advected backward in time during 11 days forced by the atmospheric fields provided by the WRF simulations. To find the origin of the moisture, specific humidity $q$ content change along the trajectory described by each particle is calculated as,

$$e - p = m\frac{dq}{dt} \tag{1}$$

where $m$ is the mass of the particle and the difference between $e$ and $p$ takes account of the increasing or decreasing ratio for the water vapour along the trajectory. From the previous equation, we can estimate the net freshwater flux over a model grid cell of area $A$ ($1° \times 1°$), summing the variation rate in specific humidity for all the air parcels ($K$) contained in the atmospheric column over that area:

$$E - P \approx \frac{\sum_{k=1}^{K}(e - p)}{A} \tag{2}$$



This methodology allows calculating the $E - P$ balance, but not evaporation ($E$) or precipitation ($P$) rates. However, we can approximate $E \approx 0$ when $E - P < 0$, since when precipitation occurs ($P > 0$), it usually greatly exceeds the evaporation rate ($P \gg E$). For this same reason, if $E - P > 0$, we can consider that $P \approx 0$. Applying the previous approximation, $E$ and $P$ can be diagnosed separately.


In order to find the moisture source regions that fed 1982 WMR catastrophic precipitations, we calculate the balance $E - P$ only for those air parcels involved in these episodes. Based on the precipitation fields provided by the WRF simulations (Figure 2 (b) and 5 (b)), for the October 1982 case we only consider those particles contained within the affected region (5°W-1°E and 37°N-42°N, Figure 1 (c)) at some point during the event (from October 19, 06UTC to October 21, 21UTC). Likewise, for the November 1982 event, we consider particles within 2.5°W-5°E and 40°N-47°N (Figure 1 (c)) from the November 6, 06UTC to November 8, 21UTC . Furthermore, to ensure that we only select those particles that contribute to precipitation, we will only select those that lose moisture $\frac{dq}{dt} < -0.06$ g(kg·3hr)$^{-1}$ over an area with outstanding accumulated rainfall rates $E - P < -2$mm(3h)$^{-1}$. Since we are only considering a subset of the total air parcels, $E - P$ cannot be seen in this case as the net surface freshwater flush, but as an indicator of where the particles contributing to the extreme rains gained or lost moisture. We can then separate the moisture uptake ($E$) from the moisture losses ($P$) by applying the approach described above. Since we are interested in the source rather than the sink regions, the $E$ field will be especially important in our analysis. Both $E - P$ (then $dq/dt$) and E fields are calculated every 3 hours and subsequently accumulated during 1, 4, 7 and 11 days prior to the precipitation events studied.

The Lagrangian methodology for the water vapour sources study exposed here suffers from two important deficiencies: (1) moisture phase changes along trajectories are not taken into account and (2) the E field represents the moisture uptake of parcels involved in the precipitation events, but those gains may not actually contribute to the events, but to a previous precipitation discharge.

Disregarding the liquid water or ice implies that clouds formation will give rise to negative $e - p$ values, when in fact that specific humidity loss are not due to a precipitation process. Likewise, part of the rain evaporates when it falls, and this will lead to positive values of $e - p$, when in fact that specific humidity gain is not due to a surface evaporation flux. The calculation of the $E$ field will be affected by this last inaccuracy. We will try to reduce this problem by taking into account only those particles that gain moisture below 1.5 Boundary Layer Height (BLH), since the water vapour uptakes in the free atmosphere will hardly be due to a surface flux. The value 1.5 is adopted to take into account the recurrent underestimation of the marine boundary layer by atmospheric models (Sodemann et al., 2008). Hereafter, to indicate those uptakes occurring below 1.5 BLH we use the subindex $_{BLH}$. In case the subindex is absent, we consider all the particles in the entire column.

An air parcel tracked back in time over a 11-day period can suffer different gains and losses of water vapour during that 125  period. Suppose for example that one of the air parcels involved in the 1982 Mediterranean rains has positive values of $dq/dt$



eight days before the event, when it was located over the Tropical Atlantic. The fact that the particle gains humidity in that area does not guarantee that this region will end up contributing to the event, because it is likely that this humidity will precipitate before that particle reaches the Mediterranean. Therefore, areas with positive values of the E field or the E-P field should not be interpreted as moisture sources but as potential moisture sources. Sodemann et al. (2008) proposes a method to avoid this

problem and to quantify the relative contribution of the sources. It is based on tracking, for a given moisture uptake, all subsequent gains and losses to know if that gained moisture reaches the study area or precipitates before it does. Once we know the uptakes that really contribute to the rain event analyzed, we can recalculate the E field, which in this case will be a more faithful representative of the moisture sources.

Using the methodology of Sodemann et al. (2008) we can define the relative contribution ($RC$) of a particular region of area $A_i$ as,

$$\%RC = 100 \cdot \frac{E|_{A_i}}{E|_{A_T}} \tag{3}$$

where $E|_{A_i}$ is the surface integral for the E field accumulated in a period of 11 days prior to the event. $E|_{A_T}$ is the equivalent but for the total domain area $A_T$.

## 2.2   Eulerian approach

The Eulerian-online method consists of a moisture tagging tool coupled to a regional or a global atmospheric model. The code of the model is modified in order to calculate new variables, called moisture tracers, which represent, for example, water vapour or cloud water coming from a desired moisture source region. This allows to estimate in great detail the relative contribution of each considered source to a given precipitation event. As mentioned in the introduction, the Eulerian tool used in this study

is the WRF-WVTs tool (Insua Costa, Damián and Miguez Macho, Gonzalo and Llasat Botija, 2017) , a moisture technique recently implemented in the WRF model version 3.8.1 (et al. Skamarock WC, 2008).

The results obtained using the WRF-WVTs for both 1982 precipitation events were already presented in Insua-Costa et al. (2018). For these two case studies they analyzed moisture coming from two-dimensional (2D) and three-dimensional (3D)

sources. To take into account moisture coming from sea surface evaporation over the Western Mediterranean, Central Mediterranean and North Atlantic, they used three different 2D sources, while to track moisture from tropical and subtropical regions they considered a 3D source in order to include both evaporation and atmospheric water transport from other neighbouring tropical regions. For each source analyzed, they run a 11-day simulation with the WRF model over a large domain of 20 km of horizontal resolution and 35 vertical levels. In the present study, the simulations of Insua-Costa et al. (2018) are used as (1)

input fields to run the FLEXPART-Lagrangian model and (2) to compare the results provided by the Lagrangian tool. Note that this strategy is especially appropriate for validating the results provided by the Lagrangian model, since both the WRF-WVTs and the FLEXPART-WRF are driven by the WRF meteorological model.





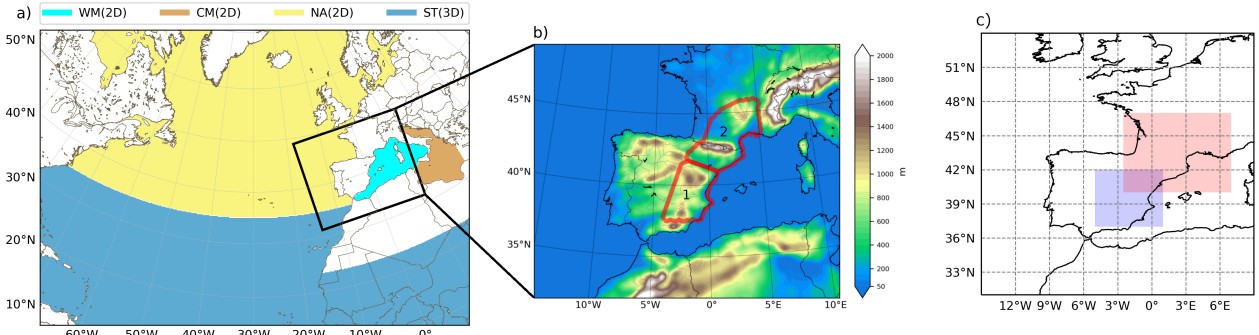

**Figure 1.** (a) Moisture sources considered: Western Mediterranean (light blue), Central Mediterranean (brown) and North Atlantic (yellow) 2D sources; and tropical and subtropical 3D source(dark blue). (b) Domain for precipitation analysis for the October (1) and November (2) event [Original figure from Insua-Costa et al. (2018)]. (c) Lagrangian particles domain selection for the October (blue) and November (red) events.

## 3 Results

### 3.1 Lagrangian moisture source diagnosis

#### 3.1.1 October event

October 20, 1982, also known as "the Tous case", was a catastrophic flooding event caused by the extraordinary rains that fell over the Spanish Levant on that day. Heavy rainfall specially affected the Valencian Community (Fig. 2b), over which a quasi-stationary mesoscale convective complex was formed, the first identified in Europe (see Romero et al. (2000), for a detailed analysis of this episode). The synoptic situation leading to such extraordinary rainfall was a classic configuration

usually affecting that region (Fig. 2a): a cut-off low near the Iberian Peninsula increases thermal and dynamic instability and, simultaneously, low pressures emerging form north Africa at lower levels organise a warm and humid flow that permanently feeds and sustains convection.

Humidity exchanges along trajectories for a small subset of particles from the previously selected are shown in Figure 3.

Only those particles that experiment a significant moisture decrease for October 20, 18-21 UTC , were considered. The high density of trajectories over the Western and Central Mediterranean indicate that much of the moisture could come from evaporation over this area. In fact, the $dq/dt$ values for air parcels crossing the Mediterranean are generally positive, indicating that they gain moisture along their paths over this sea. In addition, part of the trajectories point to a remote origin of the moisture. Some of the particles comes from the tropical Atlantic and reach the affected region after crossing the Atlantic and North

Africa. The anti-clockwise turn of the trajectories over Morocco reflects the position of the cut-off low that led to the heavy rains (Figure 2). Finally, other air parcels originated over the North Atlantic, indicating that various sources of moisture may





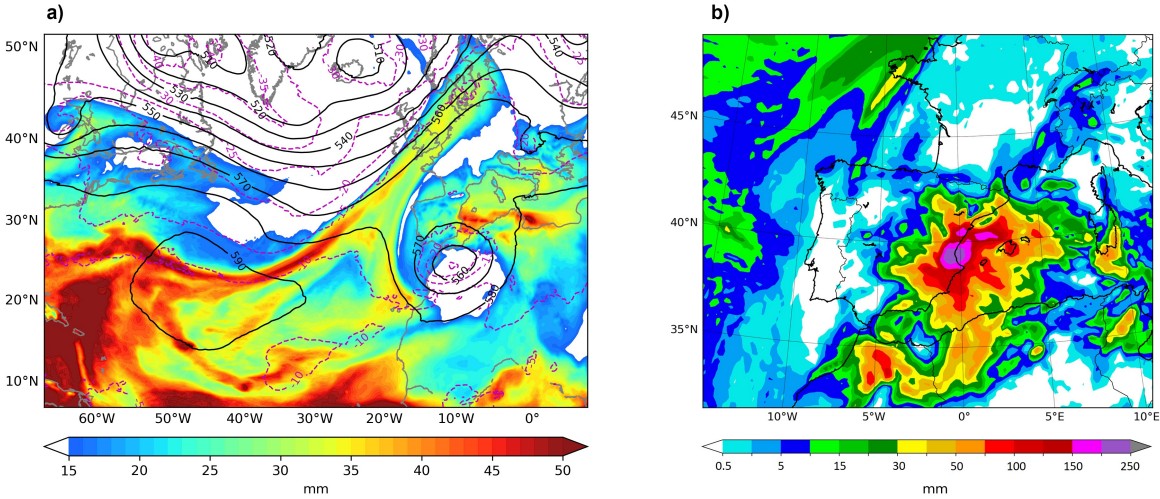

**Figure 2.** (a) Synoptic situation (from WRF simulation) on October 20, 1982 at 12:00 UTC. Geopotential height (solid black contours, dam) and temperature at 500 hPa and total precipitable water (shades,mm). (b) Simulated total precipitation (mm) from October 19 at 06:00 UT to October 22 at 06:00UTC (Insua-Costa et al. (2018)).

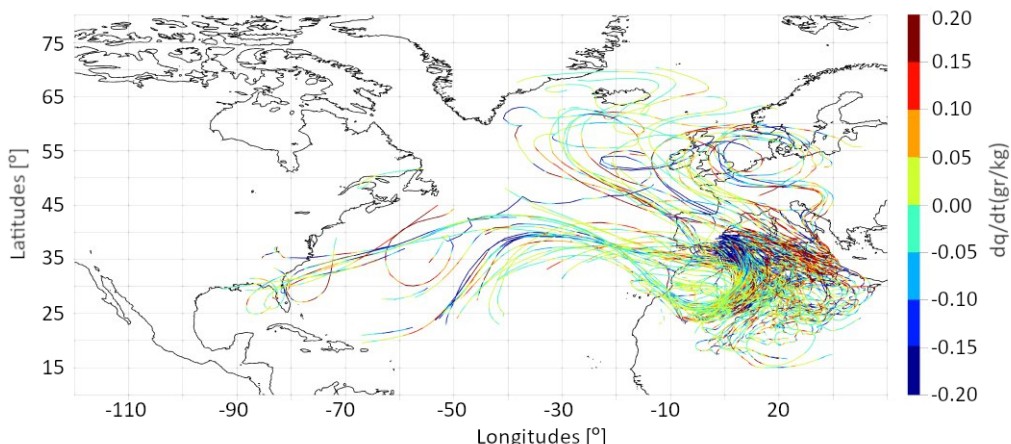

**Figure 3.** 11 day backtracking for a reduced subset of selected particles precipitating over the region of interest between 18-21 UTC on October 20. Moisture exchanges are represented along the particle path.

have contributed to the event.



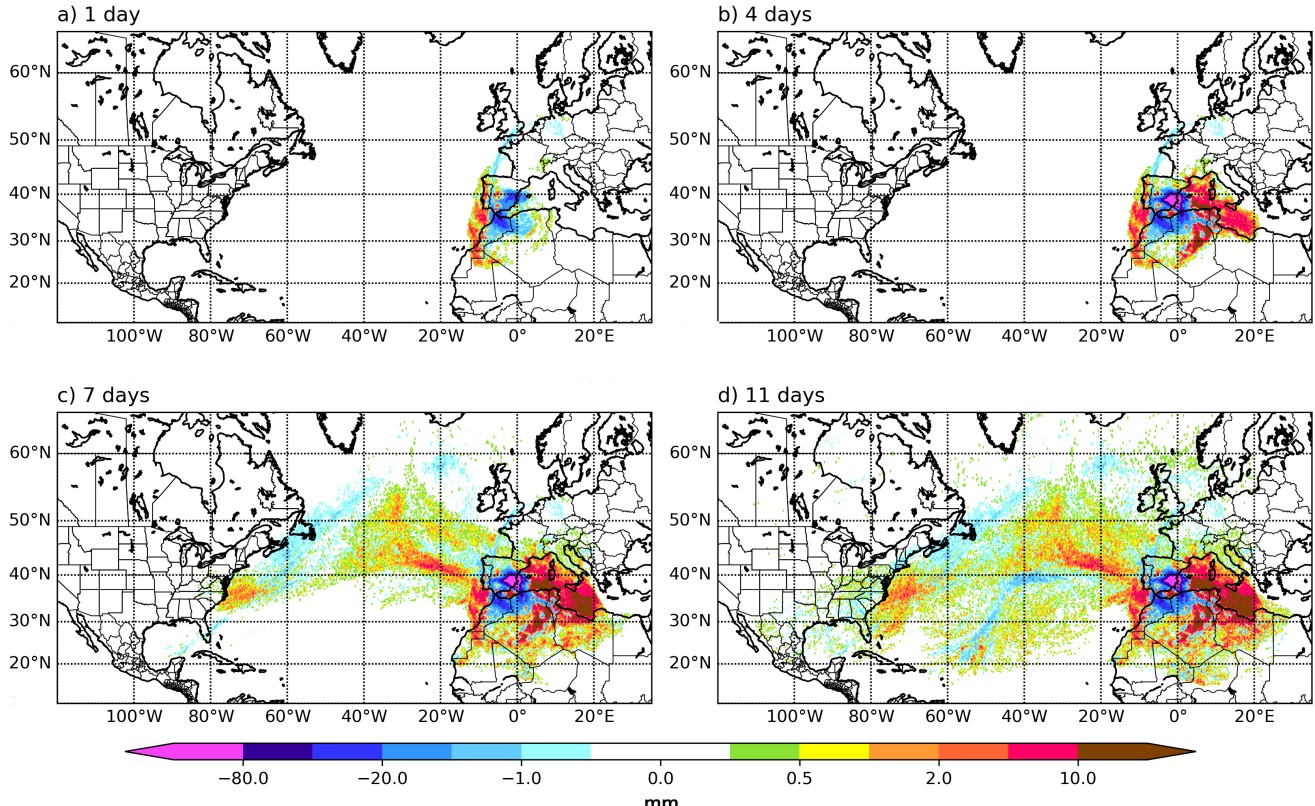

**Figure 4.** $E - P$ balance evolution back in time for 1 (a), 4 (b), 7 (c) and 11 (d) days. Particles were selected from 5°W-1°E and 37°-42°N for the 19 October 06UTC to 21 October 21UTC.

The $E - P$ balance shown in Figure 4 more clearly points the areas where the air parcels that contributed to the precipitation of the events gained or lost moisture. Specifically, we show the $E - P$ balance evolution for the time periods of 1, 4, 7 and 11 days prior to the end of the event (October 21, 06UTC) . As expected, the lowest values of $E - P$ are found over the target region during the first 3 days, which reflects the moisture discharge during the extreme precipitation event. Thus, negative values in the first days shows the area more affected by the extreme rainfall. In the first 24 hours, the highest positive values of $E - P$ are found in the eastern end of the Atlantic. Backwards in time, up to 4 days before the end of the event, the areas with positive $E - P$ values expand to much of the Western and Central Mediterranean. Therefore, these nearby areas would have fed the air parcels contributing to the event just hours before they reached the target region. In the case of the Mediterranean, $E - P$ values continue to increase up to 7 days before, indicating that not only was the contribution of evaporation in the hours prior to the event significant, but also evaporation in the previous days. In addition, a clear difference between days 4 and 7 is that positive (and also negative) values of $E - P$ appear in remote regions, mainly over the North Atlantic. These positive values extend even into the tropical and subtropical Atlantic when we calculate the accumulated $E - P$ of the previous 11 days. This indicates that the contribution of humidity from the Atlantic may have been significant. Furthermore, this humidity would





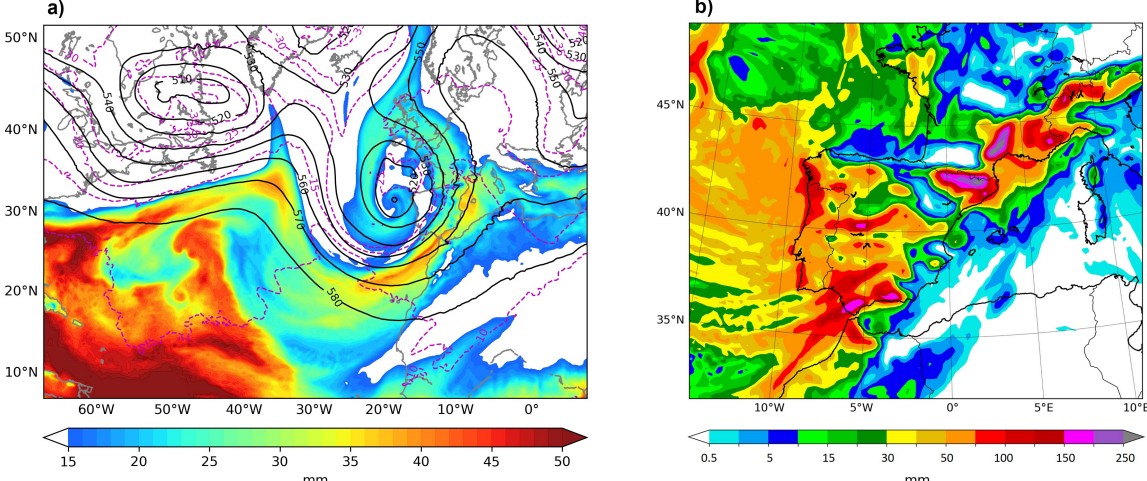

**Figure 5.** (a) Synoptic situation (from WRF simulation) on November 7, 1982 at 12:00 UTC. Geopotential height (solid black contours, dam) and temperature at 500 hPa and total precipitable water (shades,mm). (b) Simulated total precipitation (mm) from November 6 at 06:00 UTC to November 9 at 06:00 UTC Insua-Costa et al. (2018).

be older, i.e. it would have a longer residence time. One last area with positive E-P values is North Africa. Although the values at the northern coast of the continent might be realistic, the values inland are totally inconsistent since in that area evaporation is practically zero throughout the year due to the presence of the Sahara desert. Therefore, these values must necessarily be due to the limitations of the Lagrangian method. We will return to this subject below.

### 3.1.2 November event

Only a few days after the Tous case, on November 7, a catastrophic flooding occurred again in the WMR. In this case, heavy rainfall particularly affected northwest Spain, southeast France and Andorra (Fig. 5b). The atmospheric configuration that led to the excessive rainfall was very different from that occurring in October (Fig. 5a): a deep low-pressure system centred on the Atlantic coast of Galicia drove a very humid and relatively warm south-westerly flow that impacted perpendicularly against the Pyrenees and the southern face of the French Massif Central. The orographic lift in these areas gave rise to convective cores embedded in a wider area of stratiform rainfall, which in turn gave rise to persistent and occasionally very intense precipitation (see Trapero et al. (2013), for a more detailed discussion about this event).

Figure 6 shows the specific humidity content change for a subset of particles from the previously selected with significant moisture loses between 12-15 UTC November 7. In this case the particles trajectories are not as varied as in the case of Oc-





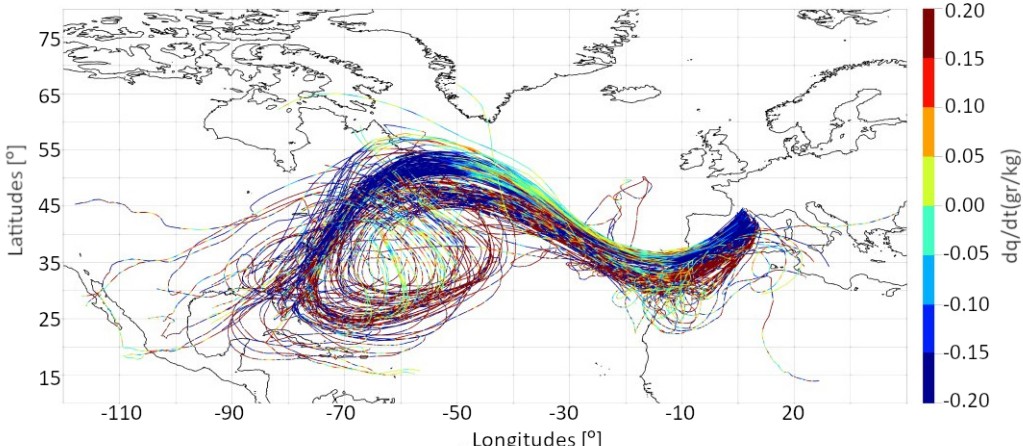

**Figure 6.** 11 day backtracking for a subset of particles precipitating over the region of interest between 12-15 UTC on November 7. Color corresponds to moisture exchanges along the paths followed by particles.

tober, but a main route is distinguished. Most of the trajectories have their origin in the western tropical Atlantic. From there the parcels would have been advected towards the north and would have crossed the Atlantic driven by the deep low-pressure
system that gave rise to the extreme rainfall event (Figure 5). In the final phase, a few hours before the particles reached the northeast of the Iberian Peninsula, the air parcels were advected northward again, entering via North Africa and southern Spain. Some of them crossed the western end of the Mediterranean before their impact. It follows that in this case the main source of moisture must be the Atlantic, while the contribution of the Mediterranean should be minor because the time that air parcels spend over this potential moisture source is scarce.


Figure 7 is analogous to Figure 4 but for the November event. In this case, the negative values of $E - P$ in the first 24 hours cover a vast region, due to the large size of the low-pressure system located off the coast of Galicia (Figure 5). Positive values of $E - P$ in the first 24 hours are only found in the Western Mediterranean. Therefore, the Mediterranean provided a last humidity recharge before the air parcels contributing to the precipitation reached the affected region. The $E - P$ field accumulated in the
4 days prior to the end of the event clearly shows the areas most affected by extreme rainfall (northeast and southwest Spain, south of France and Andorra). The positive values of $E - P$ extend towards the Atlantic and increase in the Mediterranean. Further back in time, between days 7 and 11, the highest values are found in the tropical and subtropical western Atlantic. The way the $E - P$ values are distributed is consistent with the particle trajectories (Figure 6) and also with the precipitable water field (Figure 5 (a)). In summary, the air mass in this event had a tropical/subtropical origin and was recharging its moisture
content as it passed through the Atlantic and finally through the Western Mediterranean.





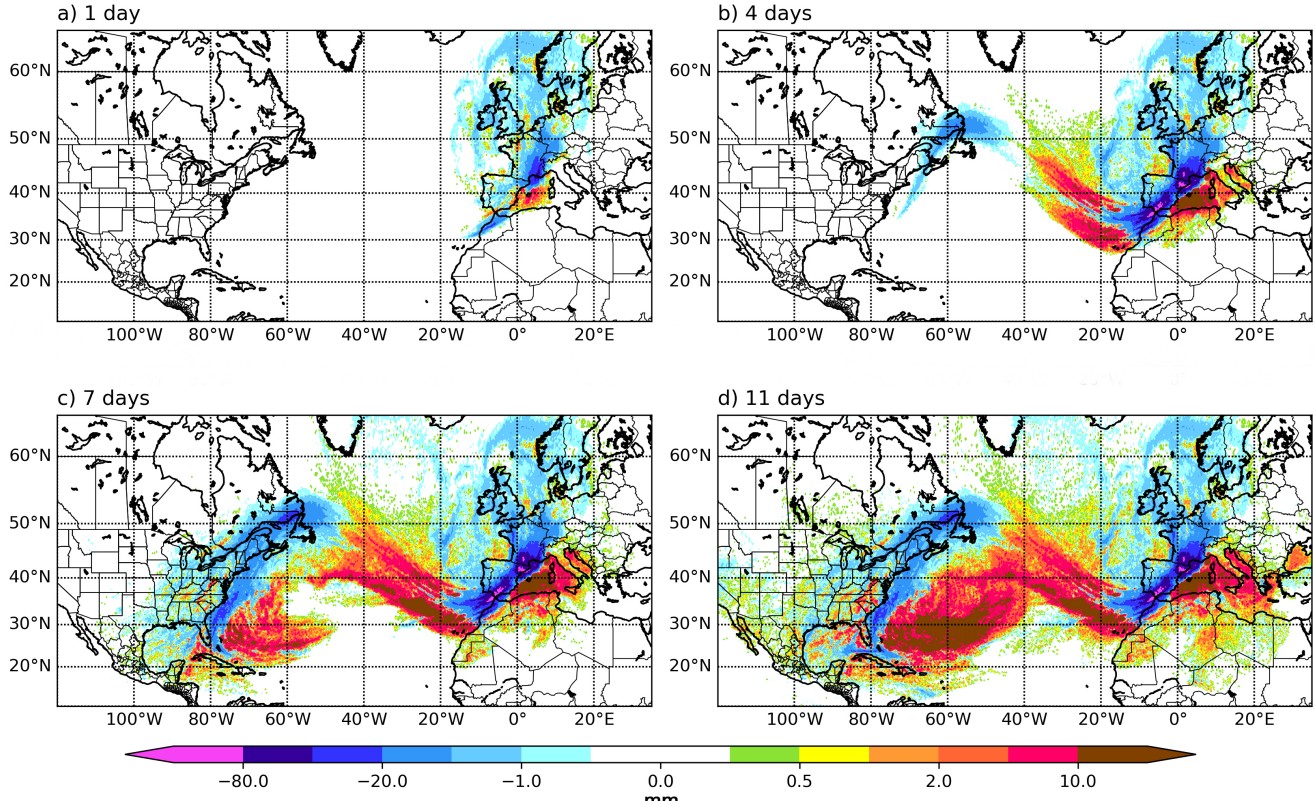

**Figure 7.** Similar to Figure 4. Balance $E - P$ evolution back in time for 1 (a), 4 (b), 7 (c) and 11 (d) days. Particles were selected from $2.5°W - 7°E$ and $40° - 47°N$ for the 06 November 06UTC to 08 November 21UTC.

## 3.2  Comparison of Eulerian vs. Lagrangian methodologies

The previous analysis of the particles trajectories and the $E - P$ field allows a qualitative analysis of the moisture sources, but not a qualitative one. The tracers Eulerian method allowed Insua-Costa et al. (2018) to calculate, for the two infamous events of autumn 1982, the percentage of precipitation coming from 4 different sources of humidity (see Figure 1): Central and West
Mediterranean sea (CMED and WMED); North Atlantic ocean (NATL) and Tropical and Sub-Tropical zones (TROP). In order to compare the results provided by the Lagrangian technique with those obtained by Insua-Costa et al. (2018) , these same percentages have been calculated using the methodology introduced by Sodemann et al. (2008), previously described.

Table 1 shows the fraction $\%RC$ (Eq. 3) of the total precipitation accumulated during the the case of October 1982 coming
from each of the four sources analyzed, calculated with both the Eulerian and the Lagrangian methods. Both methods yield similar contributions from the Western Mediterranean sea (16.81 % of Lagrangian approach versus 19.14 % of Eulerian approach) and North Atlantic Ocean (13.25 % versus 14.89 %). However, the Central Mediterranean contribution according to





| | October Event | | | | November Event | | | |
|---|---|---|---|---|---|---|---|---|
| | WMED | CMED | NATL | STROP | WMED | CMED | NATL | STROP |
| RC Lagrangian | 16.81 | 7.38 | 13.25 | 8.05 | 11.44 | 0.83 | 29.41 | 32.69 |
| $RC_{BLH}$ Lagrangian | 21.56 | 14.38 | 10.43 | 5.39 | 14.44 | 1.35 | 26.58 | 33.94 |
| RC Eulerian | 19.14 | 18.28 | 14.89 | 31.02 | 15.60 | 2.96 | 18.20 | 51.39 |

**Table 1.** Relative moisture contribution calculated for October and November cases for the Lagrangian and Eulerian simulations. $RC_{BLH}$ was calculated as in Eq.(3) bt considering particles below BLH. WMED, CMED, NATL, and STROP correspond to Western Mediterranean, Central Mediterranean, North Atlantic and South Tropics areas, respectively.

the Lagrangian method, is half of that calculated with the tracers technique (7.38 % versus 18.28 %). The most surprising difference between the results provided by both methods is found in the tropics and subtropics; while for the Eulerian method this is the main source (31.02%), for the Lagrangian method its contribution is only 5.39%. The great difference for this source finally results in that the Eulerian method assigns about 83 % of relative contribution to the four sources considered, while the Lagrangian method reports a much smaller contribution for these same areas (45.49 %).

For the November event, relative contributions are shown in Table 1. In this case, Lagrangian method estimate 74.37 % of relative contribution for the four considered sources, a very similar value to that obtained by the Eulerian methodology (88.15 %). Comparison between both approaches shows similar estimations for Central Mediterranean sea (0.83 % of Lagrangian approach versus 2.96 % of Eulerian approach) and Western Mediterranean sea (11.44 % versus 15.60 %), the less important moisture sources for this episode. Nevertheless, for the North Atlantic, the Lagrangian method results in a significant overestimation compared to the Eulerian one (29.41 % versus 18.20 %). On the contrary, the Lagrangian method underestimates the contribution of moisture from the tropics and subtropics(32.69 % versus 51.39 %).

### 3.3 Limitations and possible improvements of Lagrangian analysis

The above results show a great difference between the relative contributions obtained from the Lagrangian method and those of the Eulerian one. The biggest difference is found in the subtropical and tropical contribution. For the case of October, the Lagrangian approach underestimates this source by 74% while in the case of November this underestimation is 36%. For the moisture sources at a shorter distance from the Mediterranean region there is in general a better coincidence, but still important biases.

This shows that this Lagrangian methodology is very limited for carrying out quantitative moisture sources analysis for specific case studies. That limitation seems to affect especially the moisture contribution from long range or remote sources, as in our case are the tropics and subtropics. The increase over time of inaccuracies in particles trajectories calculations or in specific





moisture content interpolations could be behind this problem. Notably, in the November event the underestimation for tropical and subtropical moisture was substantially less than in the October case. The November episode was more "dynamic", with higher winds that would have made air parcels to travel faster along a more limited pathway. In the October events particles

travels slower trough different paths so they disperse more backward in time and spend more time in regions of weak wind. This difference could have been crucial and have made more problematic the calculations in the October case.

Therefore, the Lagrangian approach used here is more appropriate for a qualitative analysis. This qualitative interpretation of moisture sources is almost always taken from the $E - P$ field maps (Figure 4 and 7). However, we hold that, since we

are interested in moisture sources, it is more appropriate to analyze the $E$ field separately. For example, sources of moisture are usually identified from positive $E - P$ values, but an area with negative values could also contribute. For example, in the October case remarkable positive $E$ values (Figure 8 (a)) are found over the southern half of Spain and in the northern tip of Morocco and Algeria, suggesting that moisture evaporated over land could have had an important contribution. However, $E - P$ values (Figure 4) are highly negative for that area because it was the region most affected by the heavy rains (Figure

2 (b)). Something similar occurs in the November event (Figure 8 (b)), for which positive $E$ field values are found over the Iberian Peninsula, France and over the Atlantic sector north of the Canary Islands, while for these places $E - P$ is highly negative (Figure 7). Positive $E$ values in these areas, although lower than in other parts of the study region, may be especially important because of their proximity to the affected region. It should be remembered that earlier or more distant moisture gains ($E$ or $E - P$ positive) are less likely to contribute to the extreme event, as that moisture may have been lost in a previous

precipitation discharge. Conversely, the closer ones are more likely to have contributed.

Another important consideration is the unrealistic values that the Lagrangian technique used yields in some parts of the region of studied. These values are especially evident for the October case over the Sahara desert, as stated above. $E - P$ and $E$ are highly positive suggesting an important moisture gain. However is not possible that that gain comes from a surface

evaporation flux because evaporation over that area is essentially zero. This moisture increase could have been due to phase changes within the particle. Cloudiness was abundant over most of the northern countries of the African continent, so some of this water would have evaporated into the very dry Saharan lower atmosphere, leading an increase in the particles water vapour content.

Figure 9 shows the temporal evolution of the accumulated moisture increases for all particles contained in the atmospheric column over North Africa (from latitude 23°N upwards). During the first days considered, that is, between 5 to 11 days previous to the event, moisture increases are residual because of the low number of air parcels over this region. From day 3 to 5 before the event, an important amount of particles contributing to the precipitation event are already present, and the moisture gains rise. Most of these moisture uptakes take place within the BL during the central hours of the day, when the BLH is higher.

However, during the rest of the day the highest moisture uptakes occur above the BL. In fact, two days (approximately 50 hours) before the end of the event, when the maximum increase in humidity is recorded, the gains above the BL are dominant.



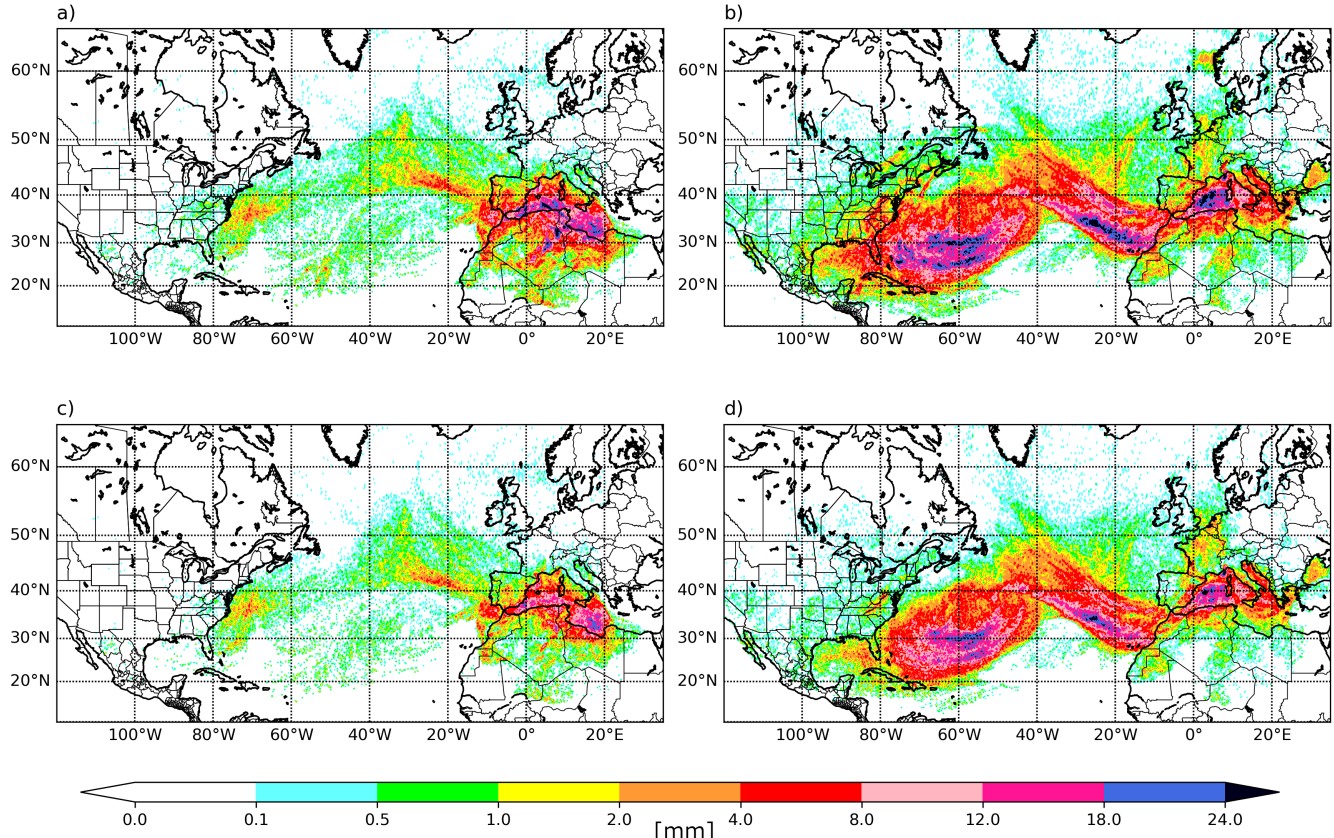

**Figure 8.** E field maps accumulated during 11 days for considered particles (a) in the entire column for October event, ($E$), (b) the same but for November event, (c) just below BLH for October event, ($E_{BLH}$) and (d) the same but for November event.

This peak could be due to the appearance of cloudiness associated with the formation of the cut-off low over Morocco (Figure 2 (a)), which in turn would lead to the appearance of phase changes above the BLH. Within the BL, gains are also important in the 3 days before the end of the event. Some of these uptakes could be real, i.e. it could actually come from a surface evaporation flux. In fact, during October 19th, significant rainfall occurred in North Africa (Figure 2), which could have increased the evaporation rate over this area the next day, when the rain had already moved to the Mediterranean. However, some of the moisture increases below the BLH may also be due to phase changes within the PBL.

As explained in the Methods section, in an attempt to reduce this problem, the $E$ field has been recalculated taking into account only the gains below 1.5 BLH. The results are shown in Figure 8. Most notably, the removal of moisture gained above 1.5 BLH clearly reduces the North African contribution in the October event (Figure 8 (c)). The $E$ field values fall across the domain, but especially in this area. In this case, the Western and Central Mediterranean stand out more as moisture sources,





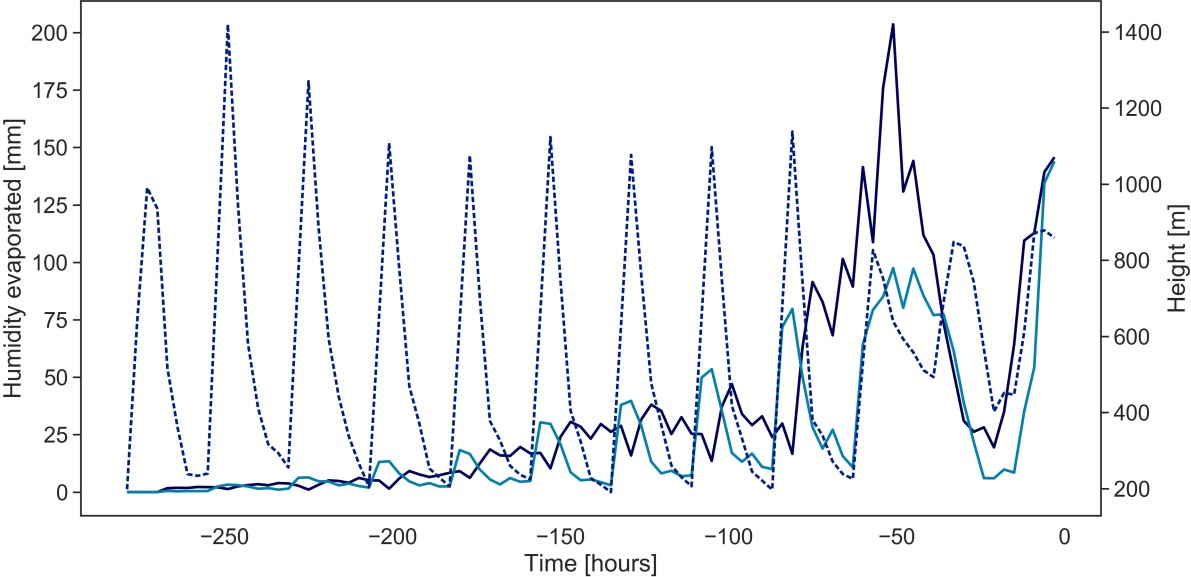

**Figure 9.** Temporal evolution of the humidity evaporated below boundary layer (light blue line), above boundary layer (dark blue line) and mean BLH (dashed blue line) for particles crossing Sahara region for the October event.

which is a more expected or realistic result than that shown in Figure 8 (a). Therefore, this new approach to calculate the $E$ field may improve the qualitative description of the sources in some cases. In the November episode, however, the reduction of

$E$ values is more homogeneous across the domain (Figure 8 (d)) and therefore the qualitative analysis that can be extracted is essentially the same.

Finally, we have also recalculated the contributions of each source considered in Insua-Costa et al. (2018) taking into account only the moisture increases within the BL. Results are also shown in Table 1 (second row). The contribution of the

Mediterranean Sea increases, as expected from Figure 8 (c). This makes the percentage of precipitation coming from the Central Mediterranean increase from $7.38\%$ to $14.38\%$, which is better adjusted to the $14.89\%$ of the Eulerian method. However, for the more distant sources, the North Atlantic and the tropical and subtropical zone, the contributions decrease and the results get worse, which makes the errors similar overall. In the case of November, however, contributions improve for all sources considered. Again, the moisture contribution of the Mediterranean rises. Thus, for example, for the Western Mediterranean the

deviation from the Eulerian method is reduced from $-4.46\%$ to $-1.16\%$. In the case of the North Atlantic the contribution drops by about $3\%$, which reduces the great underestimation that the Lagrangian method yields for this source. For the tropical and subtropical zone, the contribution increases slightly which reduces the underestimation for this source. Therefore, in general this new approach improves the results, but does not avoid the large deviations for some sources.





## 4 Summary and conclusion

The two most used techniques for the study of the moisture origin are the Lagrangian and Eulerian models. The use of one or the other can be a controversial point (van der Ent et al. (2013)), but in general it is clear that Lagrangian Models are more computationally efficient which makes them more practical. For this reason they have been widely used. However, it is important to take into account the limitations of this method for more rigorous analysis.

Here, we have compared the results of the Lagrangian FLEXPART-WRF model with those obtained from the Eulerian WRF-WVTs model for two extreme rainfall events occurred in the Western Mediterranean region in October and November 1982, respectively. The objective was to validate the performance of the Lagrangian approach, assuming that the results of the WRF-WVTs represent reality (Insua Costa, Damián and Miguez Macho, Gonzalo and Llasat Botija, 2017), which supports this assumption.


The results show that the use of the Lagrangian methodology for a quantitative study (Sodemann et al., 2008) of the moisture sources is very limited. For the nearest sources, as in our case it is the Mediterranean Sea and the North Atlantic Ocean, the results obtained with both the Lagrangian methods and the Eulerian methods have in general a remarkable coincidence. However, the contribution of tropical and subtropical areas is not adequately captured by FLEXPART-WRF, but is widely underestimated.


Therefore, we propose that the model is more appropriate for a qualitative description of the moisture origin. However, it should be noted that the model produces unrealistic values in some areas. In our case these unrealistic values become especially evident over the Sahara region during the case of October. Due to the low evaporation rate in that zone, it is impossible for it to act as one of the main moisture sources, as the Lagrangian approach suggests. We show that the humidity increases in 345 the air parcels that contribute to extreme precipitation occur mostly above the Saharan BLH, so they must be due to other processes such as phase changes of hydrometeors. The abundant cloudiness over this area on October 19th and 20th makes this hypothesis more plausible.

Therefore, Lagrangian moisture tracking tools should be further improved with the objective of being able to carry out quan350 titative analyses. The focus should be on avoiding the significant increase in errors over time, which results in the inability to correctly assess the moisture contribution from remote sources, which is often crucial in extreme rainfall events. On the other hand, liquid water and ice must be considered to improve results in some areas that are particularly sensitive to phase changes, as in our case is the north of the African continent.



*Author contributions.*  S.C. carried out the experiment and wrote the manuscript with support from D.G. and D.I.. V.P.M. and G.M.M helped supervise the project. G.M.M. conceived the original idea and D.G. supervised the project.

*Competing interests.*  Authors declare that no competing interest are presented.

*Acknowledgements.*  Funding comes from the Spanish Ministerio de Economia y Competitividad OPERMO (CGL2017-89859-R to GMM, VPM, DIC and SCG) and the CRETUS strategic partnership (AGRUP2015/02). All these programs are co-funded by the 20 European Union
ERDF. ERA-Interim data was provided by ECMWF. Computation took place at CESGA (Centro de Supercomputacion de Galicia), Santiago de Compostela, Galicia, Spain. This research is part of the HyMeX international programme.





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
