# Peer review of "Extreme precipitation events in the Mediterranean area: contrasting two different models for moisture source identification"

_Hydrology and Earth System Sciences, 2020_

## Author Comment (AC1)

**Answer to Anonymous Referee (R1) in the interactive comment.**

I found the study interesting, with a clear experimental setup and straight-forward analysis. The writing needs to be improved, I have some suggestions, but I recommend further proof-reading. I have some concerns regarding the explanation of the results and the overall conclusions of the manuscript so I recommend Major Revisions.

Thank you very much for your review and your time. We are confident that the modifications you suggest will improve the manuscript. Please find below the detailed responses to your comments.

Main concerns

My main concern is that the authors limit themselves to highlighting the deficiencies in the Lagrangian methods with little effort to propose improvements. In the abstract they state that the deficiencies are related to phase change, but the results do not provide enough evidence to support this statement. How did you come to the conclusion that phase change is the main problem? Can you quantify this? How would you propose to improve the Lagrangian methods to incorporate phase change?

In the introduction they state "the present work is intended to contribute to improving the Lagrangian analysis" but currently the authors mainly highlight deficiencies.

As such, the last paragraph in the manuscript falls short of conveying a way forward to improve the science.

We fully agree with the reviewer that our assumption that a large part of the errors found came from phase changes was based on weak arguments. The problem of omitting phase changes is well known and was argued already in Stohl and James 2004. In the cases studied, especially in the October case, we thought it made sense that the unrealistic values found in the Sahara were due to this fact, since we understood that, with cloud cover over this region in the days leading up to the event, it was very likely that some of the liquid water had evaporated on contact with the very dry Saharan boundary layer.

Motivated by the reviewer's comment, and also by Ruud van der Ent's comment, we have tried to quantify how phase changes affect the results obtained. To this end, we have taken advantage of the fact that the WRF model provides us with 6 moisture species (vapour, cloud water, rain water, snow, ice and graupel) to include the sum of all these species in the Lagrangian analysis. That is, we have repeated the calculation shown in Eq. 1: e-p=m*dq/dt, but in this case q would be the sum of all moisture species within an air parcel instead of just water vapour. Results have shown that the effect of including liquid and solid water in the model is very small. Van der Ent et al. (2013) came to this same conclusion using a completely different method.

Therefore, our hypothesis was wrong, so we have decided to make a major revision of the article. In this new version, moisture phase changes are only discussed in

section 3.3, where we show that their contribution to the errors found is small (see new Figure 9). Sentences such as "We argue that such an inconsistent contribution is associated with the fact that the Lagrangian method does not consider moisture phase changes" have therefore been removed. Instead, other possible errors have been discussed, such as those related to the convergence and divergence of humidity (see Figure 10).

Finally, we agree that our study does not offer any explicit improvement of the Lagrangian technique used, so the title of section 3.3 has been changed to "Limitations of Lagrangian analysis and possible causes". However, we have decided to keep the sentence "the present work is intended to contribute to improving the Lagrangian analysis", since we believe that our study, although it does not offer an improvement of the model code, can improve the interpretation of the results provided by the model.

Stohl, A. and James, P.: A Lagrangian analysis of the atmospheric branch of the global water cycle: Part 1: Method description, validation, and demonstration for the August 2002 flooding in central Europe, Journal of Hydrometeorology, https://doi.org/10.1175/1525-7541(2004)005<0656:ALAOTA>2.0.CO;2, 2004.

van der Ent, R. J., Tuinenburg, O. A., Knoche, H. R., Kunstmann, H. and Savenije, H. H. G.: Should we use a simple or complex model for moisture recycling and atmospheric moisture tracking?, Hydrol. Earth Syst. Sci., 17(12), 4869–4884, doi:10.5194/hess-17-4869-2013, 2013.

The main results of the Insua-Costa et al. 2018 study should be much clearer. The simulation setup, length of simulation, boundary conditions, horizontal resolution and main results should appear in a paragraph on their own before showing the results of the Lagrangian analysis.

Some of this information was already contained in the second paragraph of section 2.2 (Eulerian approach). However, following the reviewer's suggestion to include more detailed information, we have also added the boundary conditions and parameterizations used by Insua-Costa et al. 2018. The information related to the parameterizations has been included in a new table, in which we summarize the main features of the WRF-WVT and FLEXPART-WRF models.

Table 1 is arguably the most important result however, it seems insufficient to make the argument. It would be good to include a graphical display of results. Also, can you represent the results as a time series? Would this give additional insight?

Thank you for the suggestion. We have now replaced Table 1 with the following bar chart:

[Figure]

However, we believe that it is not possible to compare the results provided by the Eulerian and Lagrangian models using a time series, since the basis of each methodology is different. Specifically, the Eulerian method is not useful for creating plots similar to those in Figures 4 and 7; that is, the Eulerian method tells us where the moisture came from but tells us nothing about how many days earlier that moisture evaporated.

I was confused about the results from RC Lagrangian and RC_BLH Lagrangian. Do the latter (RC_BLH Lagrangian) use the method of Sodemann et al. 2008? If so, please clarify when you are discussing the results. Also, when you discuss the results, this is left to a last paragraph. However, it seems best to discuss the three methods together. RC and RC_BLH are very similar techniques, so it doesn't make sense to discuss them separately.

Both ratios are estimated considering the method of Sodemann et al. 2008. At the reviewer's suggestion, the definition of RC_BLH has been added to the methodology section together with the definition of RC. However, we believe it is better to discuss them separately in the text. This is because RC_BLH is calculated as an attempt to reduce the shortcomings of the Lagrangian method, so it makes more sense for it to appear in the "Limitations of the Lagrangian analysis and possible causes" section.

I think there needs to be more detailed explanation in some cases. In figure 8, what methods did you use? I am guessing these are RC and RC_BLH, but I am not sure. The same with Figure 9, what methods are you using? what is the exact domain of analysis? It is unclear to me which lines correspond to which axes.

Both figures have been eliminated in this new version. Figure 8 was eliminated at the suggestion of the second reviewer. Figure 9 was eliminated because we considered that it was no longer meaningful, since we showed that the contribution of moisture phase changes to the unrealistic values found was very small. Two new figures have been included to replace them.

Abstract Line 6: You state that these methods are "complex". Compared to what? Please read and include the following paper that will help you justify classification of the models by complexity and show another example of using WRF with water vapor tracers as the "truth" to improve other models.

Dominguez, F., H. Hu, J.A. Martinez, 2019: Two-Layer Dynamic Recycling Model (2L-DRM): Learning from Moisture Tracking Models of Different Complexity, J. Hydromet. V. 21 I. 1 DOI: 10.1175/JHM-D-19-0101.1

We have removed the word "complex" in the abstract, as suggested by the reviewer. In addition, we have added the reference to Domínguez et al. (2020) in the Introduction.

Please reference Figure 1 and explain clearly in the text what it depicts.

A clearer explanation of figures 1 (a) and (b) has been included in section 2.2 (Eulerian approach). Figure 1 (c) had already been explained in section 2.1 (Lagrangian approach).

Line 277: "Positive E values in these areas…" Doesn't this contradict the main finding that the Lagrangian technique is particularly bad for remote sources?

We do not understand why the reviewer relates this sentence to the conclusion that the Lagrangian technique is especially bad for remote sources. Here we only want to show that positive E-field values near the rain-affected area are more likely to show a moisture source region than if the positive E values were much farther away. This is because, as discussed in the paper, a moisture gain (E>0) in a very distant region is likely to be lost along the way before the air parcel gaining this moisture reaches the Mediterranean. In any case, the E-field maps have been removed from the paper at the suggestion of the second reviewer, so this discussion is no longer included in the article.

Line 325: "The two most used techniques…" This is not really true. There are MANY studies using analytical methods. Please see the Dominguez et al. 2020 reference.

Following the reviewer's suggestion, the sentence "The two most used techniques for the study of the moisture origin are the Lagrangian and Eulerian models" has been replaced by "Two of the most used techniques for the study of the moisture origin are the Lagrangian and Eulerian models". The reference to Dominguez et al. 2020 has been included in the introduction.

Minor Issues

All minor corrections have been introduced in the text as suggested by the reviewer (see version of the manuscript with changes marked). The only exception is the comment concerning line 9 of the abstract, which has not been included since we believe it is better to keep the enumeration.

---

## Author Comment (AC2)

**Answer to Obbe Tuinenburg (R2) in the Interactive comment**

I have read and assessed the manuscript. Although I like the topic of the study, I am a bit worried about the experimental set-up, because it is unclear to me which experiments are compared. Furthermore, I am concerned about the assumptions about the atmospheric moisture budget taken in the Lagrangian moisture tracking model. I realize that these assumptions have been taken in many previous studies, but I think the they impact the conclusions significantly.

We would like to thank very much the referee for his kind remarks.. Please find below the responses to your comments.

Some of the details of the experiment are unclear to me. I believe this experiment is comparing online Eulerian to off-line Lagrangian methods, which is not entirely fair. I would recommend doing the experiment with all methods in an online mode and all methods in an offline mode, so their differences can be more meaningfully interpreted. Furthermore, a lot of details about the model settings are not included and these may be important, see our work on the assumptions influence moisture tracking models: https://hess.copernicus.org/articles/24/2419/2020/ (Tuinenburg and Staal, 2020)

We do not agree that it is not meaningful to compare online and offline methods. After all, both methodologies have the same objective: the characterization of moisture sources. If both methodologies were close to "reality", both should provide similar results regardless of whether they are on-line or off-line. In fact, other authors have previously made this type of comparison (e.g. Dominguez et al.,2019).

At the reviewer's suggestion, some more details concerning the configuration of the models have been included (see for example the new Table 1). We have also restructured the methodology for a better understanding of the experiment carried out.

Dominguez, F., H. Hu, J.A. Martinez, 2019: Two-Layer Dynamic Recycling Model (2L-DRM): Learning from Moisture Tracking Models of Different Complexity, J. Hydromet. V. 21 I. 1 DOI: 10.1175/JHM-D-19-0101.1

I am worried about the assumptions regarding the atmospheric moisture budget that are used in this study but have been used in a lot of similar studies using FLEXPART. The main idea in this model is that a change in the atmospheric precipitable water along a trajectory is allocated to the total water budget at the surface (E-P), rather than its individual components E and P. As far as I can retrace, this assumption stems from the paper by Stohl (2004) on FLEXPART. It seems to be an assumption that is convenient from the atmospheric moisture budget perspective, but it becomes problematic when you actually want to allocate changes in atmospheric moisture to either E or P. I assume this approximation was warranted in the time when FLEXPART was developed when the surface fluxes (and especially E) where very model dependent and frequently used to reduce the near-surface biases of the model wrt observations. As a result, evaporation estimates were frequently unrealistic or unphysical. I would argue that at present, the surface fluxes are estimated a lot more reliably and therefore I wonder why the fluxes are not used directly, but rather the method still relies on using the total budget. I think this practice creates significant biases in moisture allocation.

We fully agree with the reviewer on this point. Changes in the moisture content of Lagrangian particles is assigned to a process of evaporation (E) or precipitation (P), when this is not always the case. In fact, Ruud van der Ent in his comments proposes a physical mechanism, convergence and divergence, which gives rise to increases or decreases in the moisture content of air parcels that has nothing to do with E or P. This paper aims precisely to quantify the inaccuracies of the Lagrangian method and discuss its possible causes, such as the one just mentioned.

Specifically on L92. For situations where E-P<0, E is assumed to be zero. I had a look at the ERA5 data to check how well this assumption holds for the domain and days considered (36N-48N, 10W-8E, over 19-21 Oct 1982 and 6-8 Nov 1982). As the authors did, I aggregated the data to 3-hourly means (from the hourly ERA5 resolution). The fraction of evaporation that occurs when E-P<0 is about 32% of the total evaporation for the domain (globally this is about 16% for these days).

We would like to thank the reviewer for his work. Taking into account the values provided by the reviewer, we have decided to remove from our article the discussion related to the E field and also the figure in which we showed it (Figure 8). Therefore, we have kept only the E-P field, without separating it.

However, we would like to clarify that we were aware that the separation of the E-P field into its two components is problematic, especially for the analysis of moisture sources in specific precipitation events. This was already stated by Stohl and James 2004. The reason why we included it, being aware of the inaccuracies involved, is because we believe that the qualitative analysis of sources from the E-P field should be complemented with the E-field. Some researchers assign only regions with E-P>0 as moisture sources. But this need not always be the case. Areas with negative E-P could also have contributed to the precipitation accumulated in the event studied. The E-field does not suffer from this problem, which can lead to misinterpretation of the results.

Stohl, A. and James, P.: A Lagrangian analysis of the atmospheric branch of the global water cycle: Part 1: Method description, validation, and demonstration for the August 2002 flooding in central Europe, Journal of Hydrometeorology, https://doi.org/10.1175/1525-7541(2004)005<0656:ALAOTA>2.0.CO;2, 2004.

Regarding the assumption of the precipitation events, on L103, only moisture is allocated when E-P < -2 mm per 3h. Again, I had a look at the ERA5 data for the domain and days. The precipitation events for which the condition is true only represent 75% of the precipitation for the domain (globally this is only 47% for the days considered).

I think that these fractions of the evaporation and precipitation events missed is significant. Depending on how these E and P events are distributed compared to the cases studied here, the results will probably be affected quite a bit. (I realize the present study is done based on WRF simulation, which are different than the ERA5 reanalysis, but I would not expect these fractions to be very different.)

Thresholds similar to E-P < 2mm/3h have been used by other authors in the past (e.g. Stohl et al.,2008). This assumption is only used to select the air parcels that contribute to the heavy precipitation events within the target area (Figure 1c), so we see no inconsistency in this point.

Stohl, Andreas, Caroline Forster, and Harald Sodemann. "Remote sources of water vapor forming precipitation on the Norwegian west coast at 60 N–a tale of hurricanes and an atmospheric river." Journal of Geophysical Research: Atmospheres 113.D5 (2008).

L 59-60: This was not the main conclusion of Van der Ent 2013. The main conclusion of that paper was that both methods have differences in moisture flow representation compared to the on-line tracking. The main problems occur in locations with a lot of vertical variability in the horizontal integrated moisture flow, such as monsoon areas with strong flows at the surface and return flows higher up in the atmosphere. The Eulerian model in that study initially considered the vertical integral of the horizontal moisture flow, leading to underestimation of horizontal moisture flow in situations where the flows are opposite in the lower and upper atmospheric levels (thus cancelling each other). Based on this study (and a lot of other work), the WAM model was adapted to use two layers in the vertical.

Thank you for your comment. The introduction has been modified and the sentence "where the evaporated moisture from Lake Volta (in West Africa) was tracked until it precipitates, concluding that the Lagrangian method leads to inaccuracies in the calculations in the presence of strong wind shear" is no longer included in the text.

L360: How exactly was the ERA-Interim data used in the present study? I did not find any other mention in the manuscript rather than in the acknowledgements.

The ERA-Interim was used as initial and boundary conditions for the Eulerian WRF-WVTs model. Since the results provided by this model had already been published in another study (Insua-Costa et al., 2019), we have decided to remove this database from the acknowledgements.

Insua-Costa, Damián, Gonzalo Miguez-Macho, and María Carmen Llasat. "Local and remote moisture sources for extreme precipitation: a study of the two catastrophic 1982 western Mediterranean episodes." Hydrology and Earth System Sciences 23.9 (2019): 3885-3900.

---

## Author Comment (AC3)

**Answer to comments by Dr Ruud van der Ent – Delft University of Technology**

After seeing part of this work presented at the EGU GA in 2020, I am happy to see this work in written form. The comparison of different moisture tracking models is important and timely. In this paper, the authors highlight several errors in moisture tracking arising from a Lagrangian offline method FLEXPART-WRF in comparison to an Eulerian online method WRF-WVT. I have some comments that I think the authors should be able to take into account in a revised version of their manuscript.

We would like to extend a special thanks to Ruud van der Ent for his brilliant comments, which have been of great help in improving the article.

The authors present their results as Eulerian vs. Lagrangian, but I think this is misleading. In fact there are many more differences between the methods as indicated roughly in the table I made below. I think it would be good if the authors expanded/improved this table and used it in their paper. Also, I would actually suggest adjusting the title, because these findings cannot be generalized to all Eulerian or all Lagrangian models. For example, the moisture tracking method I developed myself is Eulerian offline and not very computationally demanding.

Thank you for the time invested in the table. We think it is a very good idea and we have included it. We also agree that we have not been careful enough about the distinction between the different Eulerian and Lagrangian methods. Indeed, the results obtained cannot be extended to all types of Lagrangian and Eulerian methods, so we have modified the text (also the title) to try to improve this weak point of the article.

On line 13-14 the authors conclude "We argue that such an inconsistent contribution is associated with the fact that the Lagrangian method does not consider moisture phase changes." However, I do not think that the phase changes play a major role. In Van der Ent et al., (2013) we found that the effect of phase changes on moisture tracking is really minor and does not significantly effect the patterns of the moisture tracking.

Indeed, our hypothesis that the unrealistic values found in the Sahara were due to phase changes was wrong. Motivated by your comment and also by the first reviewer's comment, we have tried to quantify how phase changes affect the results obtained. To this end, we have taken advantage of the fact that the WRF model provides us with 6 moisture species (vapour, cloud water, rain water, snow, ice and graupel) to include the sum of all these species in the Lagrangian analysis. That is, we have repeated the calculation shown in Eq. 1: e-p=m*dq/dt, but in this case q would be the sum of all moisture species within an air parcel instead of just water vapour. Results have shown that the effect of including liquid and solid water in the model is very small. Specifically, the E-P field values change by about 4% on average (absolute) when liquid and solid water are included (see section 3.3 of the article).

Yet, of the differences mentioned in the table above, the key problem with FLEXPART-WRF in my understanding is the combination of the Lagrangian moisture pathways AND evaporation attribution by E-P balance. As Obbe Tuinenburg already pointed out in his review this does not work very well because E and P can be concurrent during the same time step. But there is also another issue, which

surprisingly has rarely been mentioned, namely the fact that the E-P balance in a Lagrangian framework is neglecting convergence and divergence in the atmosphere. Suppose you have a grid box in a Eulerian sense and convergence takes place equally from all sides, then the volume in the grid box increases, in a Lagrangian setting a parcel exactly at the center of this grid box stays in the same place, but also its volume increases. Now, in the E-P backtracking method, the result would be an attribution to evaporation, but this is not what happened in reality. In such a way you can obtain moisture gains and lossed along the pathway with E and P both being 0. It was already noted by Stohl & Seibert (1998) that specific humidity fluctuations along a trajectory may be entirely unphysical, and Stohl and James (2004), who evaluated the FLEXPART methodology, found that when FLEXPART is used to evaluate E and P separately, evaporation is highly overestimated. In my view, this is a more logical explanation for the wrong attribution of moisture sources over the Sahara than the issue of phase changes (e.g. also discussed in lines 281-289).

We have to acknowledge that we had not been aware of this problem. We agree that this deficiency is probably to a large extent the cause of the unrealistic values found. In fact, we have included a new figure (Figure 10) outlining this problem.

A last suggestion I would like to make is that the figures now mostly just show the FLEXPART-WRF results, whereas it would in my opinion be more informative when the results of WRF-WVT and FLEXPART-WRF would be presented next to each other (i.e. a spatial version of table 1).

We have not been able to straighten out this point because the approach of the two models is totally different so we think it is not possible to make a comparison of the two methodologies in a "spatial" figure.

In conclusion, the findings of this study are important and should present a clear warning to anybody that uses the E-P method for attributing evaporative sources as the authors show that leads to major errors and unrealistic results. The authors made a fair comparison between a golden standard online method WRF-WVT and thus have all the right to be even more outspoken against the use of attributing evaporative sources based on E-P and I hope they bring across this point more strongly in a revised manuscript. Yet, they should be careful in their semantics as the conclusions may not apply to just any other Lagrangian or Eulerian method.

Thank you again for your positive remarks.

**References**

Stohl, A. and James, P.: A Lagrangian analysis of the atmospheric branch of the global water cycle: Part I: Method description, validation, and demonstration for the August 2002 flooding in central Europe, J. Hydrometeorol., 5(4), 656–678, 2004.

Stohl, A. and Seibert, P.: Accuracy of trajectories as determined from the conservation of meteorological tracers, Q. J. R. Meteorol. Soc., 124(549), 1465–1484, doi:10.1256/smsqj.54906, 1998.

Sodemann, H., Schwierz, C. and Wernli, H.: Interannual variability of Greenland winter precipitation sources: Lagrangian moisture diagnostic and North Atlantic

Oscillation influence, J. Geophys. Res. D Atmos., 113(3), D03107, doi:10.1029/2007jd008503, 2008.

van der Ent, R. J., Tuinenburg, O. A., Knoche, H. R., Kunstmann, H. and Savenije, H. H. G.: Should we use a simple or complex model for moisture recycling and atmospheric moisture tracking?, Hydrol. Earth Syst. Sci., 17(12), 4869–4884, doi:10.5194/hess-17-4869-2013, 2013.

---

## Author Response (AR2)

I have read and assessed the revised manuscript. While the manuscript has improved, I still have some criticism on the current manuscript that should be addressed before publication.

Thank you very much for your review. We believe that the modifications you suggest will improve the manuscript. Please, find below the responses to your comments.

1. I am still unconvinced by the experimental setup, table 1 makes this clearer; there are many differences between the two models tested that could all lead to the differences found. Nevertheless, the authors all attribute this to the difference in model structure (Lagrangian vs Eulerian). My suggestion would be to do an experiment in which only one aspect of the model setup is changed to test the effects of individual aspects.

We understand the reviewer's concern on this point. However, to our knowledge, the proposed experiment is totally inaccessible. As the reviewer says, there are many differences between the two methods used and not all of them are related to the structure of the model. For example, table 1 specifies that one of the models takes phase changes into account and the other does not, which has nothing to do with the model structure. However, it is impossible to isolate some individual aspects of the model to assess their effect. In other words, it is not possible to match the two columns of table 1, except for the first row. For example, the Lagrangian methodology used inherently works in a backward direction, and it is impossible to match it to the Eulerian methodology. Even if such an experiment were possible, we are not sure it would be necessary; here we are comparing a recently coupled Eulerian tagging tool to the WRF model against a widely used Lagrangian approach, and we are interested in the overall differences (with all their aspects) between the two models. A similar comparison has been made by other authors in the past (van der Ent et al., 2013).

We consider that there is not really a problem with the experimental set-up, but a problem of language. Perhaps we have overused the terms Lagrangian and Eulerian to refer to the methods used. As the reviewer points out in the following comment, there are many different Lagrangian methods, and we should have used a more concrete vocabulary. We have tried to correct this point (see next reply), and in this way we hope to make it clear in the text that the differences found need not be exclusively due to the difference in model structure.

Van der Ent, R. J., et al. "Should we use a simple or complex model for moisture recycling and atmospheric moisture tracking?." Hydrology and Earth System Sciences 17.12 (2013): 4869-4884.

2. In many parts of the text, the results are still generalized to Lagrangian models or methodologies. This is problematic, as it gives the impression that the problems you diagnose are valid for all Lagrangian methods. However, many Lagrangian approaches have completely different assumptions. This was also indicated in the previous review, and by other reviewers. I suggest to change all mentions of Lagrangian model(s) to FLEXPART-WRF.

In the previous revision, we had already tried to solve this problem by adding the following clarification at the beginning of the Methods section: "so from now on, when we refer to the Lagrangian or Eulerian approach, we will be referring to the specific methodology explained here". However, we agree that in some parts of the text this generalisation may still confuse

the reader. Therefore, following the referee's suggestion, we have used the term FLEXPART-WRF instead of Lagrangian model when we refer to the tool used. In addition, we have differentiated between FLEXPART-WRF, E-P balance and the Sodemann et al. (2008) methodology (see second paragraph of Section 2.1); FLEXPART-WRF deals with the dynamical part (particle trajectories) while the E-P and Sodemann et al. (2008) methods deal with the hydrological part (moisture sources) of our analysis. Nonetheless, the use of the term "Lagrangian" has been retained in some parts of the text where it is considered necessary and not misleading.

3. The Figure 10 that is now added to the text is nice, and the simulation of convergence and divergence is problematic if you consider only a limited number of parcels. In our paper on the sensitivity assumptions of moisture tracking models

(https://hess.copernicus.org/articles/24/2419/2020/ (Tuinenburg and Staal, 2020)), we discuss the effect of the number of parcels on the simulation quality. You could have had a look at that and used it in your discussion around Figure 10.

Thank you for your recommendation. We have included this reference as part of the discussion of Figure 10.

4. The problem with the E-P balance is not addressed properly to my liking. The authors have added to the results section that the positive E-P values based analysis is not entirely certain (L220-230 of the track-changes manuscript). Given the large fractions of moisture not taken into account by taking the assumptions (see values in my earlier review), I would have expected some clear language.

To clarify this point we have added the following sentence in the middle of paragraph L220-230: "a preliminary analysis shows that a significant fraction of the total evaporation occurs when E-P<0 (see https://doi.org/10.5194/hess-2020-651-RC2)". We have taken the liberty of referring directly to the previous reviewer's comment because we think it is the easiest way for the reader to understand the concept we are trying to convey here.

**Answer to Ruud van der Ent in the Interactive comment on "Extreme precipitation events in the Mediterranean area: contrasting two Lagrangian and Eulerian models for moisture source identification"** by **Sara Cloux et al.**

I apologize for submitting this review only on the last day of the deadline. I think the authors did an outstanding job in improving this paper.

We thank the reviewer again for his positive comments.

I thank the authors for giving reference to my ideas about convergence/divergence as possible sources of error for the Lagrangian tracking in combination with moisture source identification based on E-P. However, instead of "personal communication" it would be more traceable if they cited the comment on the original manuscript instead: https://doi.org/10.5194/hess-2020-651-CC1

We have added the reference to the previous comment as suggested by the reviewer.